# Endometriosis, an Ongoing Pain—Step-by-Step Treatment

**DOI:** 10.3390/jcm11020467

**Published:** 2022-01-17

**Authors:** Sylvia Mechsner

**Affiliations:** Clinic for Gynecology, Endometriosis Center Charité, Charité—Universitätsmedizin Berlin, Augustenburger Platz 1, 13353 Berlin, Germany; sylvia.mechsner@charite.de; Tel.: +49-030-450-664866

**Keywords:** endometriosis, adenomyosis, chronic lower abdominal pain, pelvic pain, dyspareunia, spinal hyperalgesia

## Abstract

Endometriosis is a disease that is becoming more and more challenging for the medical community. The current therapeutic concepts (surgical therapy and/or hormonal therapies) often do not lead to sufficient pain control, and late diagnosis and high recurrence rates mean that women affected by the disease can suffer for decades before receiving proper treatment. Although the introduction of certified endometriosis centers has created contact points for surgical therapies performed by endometriosis experts, these centers are not sufficient to offer the affected patients the all-encompassing long-term support they need. In recent years, new findings regarding the pathogenesis and correlations of the pain phenomena caused by endometriosis have shown that conventional therapy strategies are not adequate and individual long-term concepts must be developed. Not only can endometriosis cause nociceptive pain, but it can also lead to a nociplastic reaction with central sensitization. Hence, aside from the classic cyclic complaints, patients increasingly suffer from atypical pain. Due to the high number of affected patients who are treated inadequately, it is necessary for gynecologists in private practices to become familiar with multimodal treatment concepts since they are the central point of contact of their patients. The following article will provide an overview of treatment strategies for chronic symptomatic endometriosis.

## 1. Introduction

Endometriosis (EM) is a widely unrecognized gynecological disease. When affected women talk about their pain, their social environment (family and school), but also their pediatricians, family doctors and, above all, gynecologists tend to simply accept it—even though this disease often lasts throughout the entire reproductive phase of their lives and in most cases causes complaints well before their 20th birthday [1]. Endometriosis must be regarded as a chronic pain disorder with a high recurrence rate, even after surgical removal [2,3]. When comparing endometriosis with other chronic diseases that are equally common, such as diabetes mellitus, it is shocking to find that the general knowledge about diabetes among people on the street is very different from the general knowledge regarding endometriosis. Not only do people know that diabetes mellitus is a disease, but it is also common knowledge that it has to do with an increased blood glucose level that can lead to serious organ damage and can even be fatal, if left untreated [4]. It is also well-known that education and making changes to one’s lifestyle can prevent most of the damage. Everyone is aware that diabetes patients need special training and that their environment such as family/school/workplace also needs to show consideration. However, this disease hardly causes any severe symptoms; but, we have evidence: the high level of blood glucose that can be measured. There is a therapy and a drug (insulin), which can be injected. The causal relationships are easy to understand. In the case of endometriosis, this is very different: When young women experience menstrual pain and lower abdominal pain, there is little possibility of confirming the diagnosis, since changes in the organs are initially hardly or not at all visible and invasive diagnostics should not always be the first choice. After all, not all young women with menstrual pain can or should undergo laparoscopy right away. Another aspect seems to be that menstrual pain is considered to be quite normal. However, what does “normal” menstrual pain mean? Is it normal to feel a bit unwell and tired, but with 1–2 ibuprofen 400 mg you can manage your daily routine without pain, including physical exercise? Or is it still normal for girls to collapse, be in pain and confined to the bed with nausea/vomiting or diarrhea, despite taking painkillers? It is true that pain is a subjective phenomenon and that it is not easy for a doctor to assess it. If the examination findings are unremarkable, it is certainly not easy for a gynecologist to make a serious diagnosis such as endometriosis. Still, in the vast majority of cases, affected women report that they have been complaining about their symptoms for an average of 10 years, that in more than 60% of the cases, the symptoms occurred before the age of 20, and that they had visited up to 10 different physicians before a diagnosis was finally made [1,5]. Until then, they received inadequate and inconsistent therapies. To this date, a conclusive diagnosis is only made after surgery. This surgery, however, is often not very well-planned or thorough, but only a histological confirmation without full remediation. If subsequent therapy concepts are inconsistent as well, this can lead to a complex pain problem which is very difficult to understand pathophysiologically. It is alarming that by now, several scientific papers have been published proving the insufficient care and treatment of endometriosis patients in gynecological practices [6]. A major problem in outpatient care is, of course, that this condition is not adequately represented in the billing system. Why is there, for example, no billing code for a medical pain history related to endometriosis? A Berlin practice specializing in endometriosis has calculated that the amount refunded for an endometriosis patient is only around 48 Euros [7]. Quite obviously, this does not in any way reflect the time and effort required for thorough medical history/examination/ultrasound/consultation and developing a therapy concept. However, is this really why the disease and its symptoms are ignored, or is this rather due to a lack of knowledge concerning the management of this disease?

It is becoming more and more apparent that the results of conventional (surgical and hormonal) therapies are not satisfactory in the long run [8]. While surgical therapy was considered the main intervention for many years, it must now be recognized and accepted that repeated surgery does not guarantee long-term improvement concerning the patient’s desire to have children [9] nor regarding pain reduction. This has to do with the pathophysiological processes of pain mechanisms. In her review articles, Mechsner [10,11] details the different levels of pain genesis (Figure 1). In the beginning, nociceptive pain (dysmenorrhea and cyclic lower abdominal pain) is the main symptom, while over time nociplastic processes also lead to a lowered pain threshold accompanied by spinal hyperalgesia and pelvic floor dysfunction, which then often results in dysuria, dyschezia, dyspareunia and chronic pelvic pain. In addition, changes in the inflammatory processes and the innervation within lesions can lead to neurogenic inflammation, which can cause hormone-independent, acyclic lower abdominal pain. Given these complex pain mechanisms and the realization that repeated surgery does not produce the desired permanent effect, the way endometriosis is managed has changed fundamentally [12]. Early diagnosis and the initiation of a consistent therapy are essential. This requires a thorough medical history, intensive gynecological palpation (including the pelvic floor muscles), detailed diagnostics (exclusion of organ dysfunction), followed by an early initiation of drug therapy. Primary surgical diagnosis is not necessary as long as there is no organ destruction, symptoms are well-managed and the patient currently has no desire to have a child. Surgical therapy can then be included in the overall treatment plan in such a way that the intervention shortly precedes the realization of the patient’s desire to have children, thus making the most of the often-positive effect of remedial surgeries. Of course, organ destruction or persistent pain during hormonal therapy continue to be indications for surgery independent of the desire to have children. This leads to multimodal concepts for the treatment of chronic endometriosis patients. The usual course of action in cases of persistent pain or an unfulfilled desire to have children is to arrange for surgical therapy to diagnose endometriosis. Subsequently, a progestogen therapy is usually recommended, preferably a progestogen monotherapy [13], but also combined oral contraception (COC) could be used [14]. This will certainly work well for the majority of patients. However, what to do if the patient continues to complain, if the hormones are not well-tolerated, if the suffering continues? This is when many gynecologists are faced with a dilemma. The Endometriosis Center at the Charité specializes in the treatment of chronic endometriosis pain patients. Thanks to the possibility of treating patients not only surgically, but also using conservative therapy, we were able to develop strategies for addressing pain on different levels. The aim here is to share empirical data introducing a possible course of action (step-by-step approach), which will need to be scientifically proven in the future.

As a first step, we wanted to understand the pain profile of patients diagnosed with and treated for endometriosis. When we analyzed the patients, we found that pain was the main reason they had come to the clinic (about 90%). Half of the women were already on hormonal therapy. Of these, 41.6% reported that their pain continued to occur “cyclically”. Further analysis revealed that these patients also bled during hormonal therapy, and that they perceived the associated pain to be cyclic. In contrast, 55.5% of the patients reported their pain to be acyclic. In patients not receiving hormonal therapy, cyclic pain was predominant (58.4%), yet even in this group, 44.4% reported suffering from acyclic pain. The median pain score on the VAS was 6. Half of all women reported other types of pain, such as dyspareunia, dysuria and dysuria [15]. This indicates a significant level of suffering.

## 2. What to Do with Pain-Ridden Endometriosis Patients

In this kind of situation, the pain problem should be assessed via an in-depth medical history. It is important to determine whether the pain could be caused by other factors or whether it might be related to surgical problems. Then, a thorough gynecologic and ultrasound examination should follow. Once organ destruction is ruled out, the working hypothesis is that the chronic pain is related to endometriosis. Figure 2 shows a practical step-by-step approach, indirectly describing the pathophysiological basics of pain. Pain due to endometrial lesions (release of pain mediators and activation of nociceptive nerve fibers) can be treated hormonally and surgically, as well as using analgesics (steps 1–2–3). The continuous pain, however, leads to more and more nociplastic changes with central sensitization, which means that the pain threshold is lowered. Incorrect and relieving postures can cause muscle cramps, and the chronic inflammation is frequently accompanied by intestinal problems. There are several complementary measures that can be applied to relieve the patients’ pain, such as physiotherapy, osteopathy, acupuncture, dietary changes, etc. (steps 4–5). As a chronic pain disease, endometriosis is a considerable source of stress. When dealing and coping with the disease, patients should also receive psychological support (step 6). Eventually, a severe chronic pain syndrome may develop. In this case, inpatient complex therapy might also be beneficial (step 7).

## 3. Practical Course of Action—Examining the Individual Steps

### 3.1. Hormonal Therapy

Since endometriosis is essentially a hormonal disease, hormonal drug therapy is a basic therapy. Thus, it is important to check whether the patient is on hormonal therapy and, if so, whether it is sufficient. If not, hormonal therapy should be initiated.

No hormonal therapy: It is important to know why this is the case. If the patient wants to have children, for example, hormonal therapy is not appropriate, since it will only waste time. In this case, other conservative multimodal strategies should be offered. However, the desire to have children should then be pursued and hastened accordingly, and if necessary, the patient should be referred to a fertility center. There are times, however, when a “respite from pain” may be beneficial to the patient. This option should be discussed if the patient is in great distress. If the patient is in severe pain, hormonal therapy may be prescribed for 3–6 months. This relieves the patient so that she can recover. However, these days, it is not uncommon for patients to deliberately turn down hormonal therapies A current trend shows that patients are reluctant to accept hormonal therapies. More and more women are rejecting synthetic hormones because they believe that they are unnatural and thus bad for the body. However, since endometriosis is a hormone-dependent disease, which can develop progressively, but can also lead to increasingly severe pain if left untreated due to central sensitization, the patient should receive objective counseling and advice. This might change the patient’s attitude towards hormonal therapy. The benefit of bioidentical progesterone is also often discussed in this context: these can be taken concomitantly and are often perceived as helpful, but there are currently no studies available, and this recommendation is based on patients’ reports. However, if hormones are not well-tolerated and have adverse side effects, one needs to think about alternatives. Although international guidelines recommend COC as a first-line therapy, I agree with Casper that progestin mono-preparations (POP) should be the better first-line option [13,14]. However, when taking dienogest (Dng), 5% of women may experience depressive moods. In general, patients who are on hormonal therapy frequently report mood changes. In this case, a small transdermal administration of estrogen might be an option. In individual cases, the vaginal application of Dng was found to be better-tolerated. Other alternatives include desogestrel, chlormadinone or drospirenon (all may need to be given at higher doses to prevent bleeding). Duphaston might also be an option. Combined oral contraceptives (COCs) are often better-tolerated, so they could also be worth trying. At any rate, local hormonal therapy with levonorgestrel (Lng) should be pointed out to at least relieve uterine-associated pain. In case the patient has simply not been offered/recommended hormonal therapy, this is the right time to initiate it.

Hormonal therapy, but bleeding: As described by Brandes et al., even light bleeding is often painful when a patient is on hormonal therapy [15]. Thus, the bleeding should be significantly reduced. The goal should be to achieve therapeutic amenorrhea. This might mean switching from POPs such as desogestrel to Dng, which has a strong antiproliferative effect on the endometrium. If the bleeding persists while on Dng, the dose can be doubled (1-0-1) for seven days. If there is no response, withdrawal bleeding should be induced. It is also recommended to sonographically assess the thickness of the endometrium as well as the functioning of the ovaries, since there might be a functional cyst with a mucous membrane. It is advisable to actively involve the patient in this, as she will be able to handle her pain better if she understands where it comes from and it will in general be easier to patiently wait for an improvement. Understandably, when the pain persists, hormonal therapy is less likely to be accepted and more likely to be discontinued. COCs should be given non-stop. Here, too, withdrawal bleeding is usually painful. If the pain results from adenomyosis, local hormonal therapy with Lng can also be effective. If the bleeding continues, Lng can also be combined with an oral POP.

Hormonal therapy/no bleeding/persistent pain, in most cases acyclic: In this case, switching from COCs to POPs (Dng) should be attempted. When taking COCs, the cumulative estrogen level exceeds 50 pg/nl, which means that the value is in the active range of proliferation. With POPs, the estrogen level usually remains within the therapeutic window. Although both Dng and Gonadotropin analogues (GnRha) achieve comparable levels of pain reduction during first-line therapy [16], in the context of chronic pain, GnRha are superior to Dng and should thus be used as second-line therapy (with add-back hormonal replacement therapy (HRT), e.g., tibolone or transdermal estrogen combined with an oral or vaginal POP). This often leads to a significant reduction and sometimes even complete freedom from pain. In the future, the orally available GnRH antagonist elagolix, which has not yet been approved in Germany, will hopefully also be available. It can be obtained via the internet, but it is very expensive (1050 Euros for 28 tablets). Relagolix, another orally available GnRH antagonist, is still approved for the treatment of fibroids and might also be available for the treatment of endometriosis in the future. If the patient shows signs of improvement after three months, the therapy should be extended to six months. There are some patients who can be treated with GnRha only and—with appropriate add-back HRT—can take it for a longer period of time. Furthermore, in case of ongoing pain, also under GnRha, aromatase inhibitor treatment (with or without progestin, COC or GnRH analog), significantly reduced pain compared with GnRH analog alone [17]. Furthermore, in case of ongoing pain, also under GnRha, aromatase inhibitor treatment (with or without progestin, COC, or GnRH analog, significantly reduced pain compared with GnRH analog alone [18].

Reminder: In order to evaluate the success of hormonal therapy, the patient MUST be free of bleeding, otherwise it CANNOT work!

### 3.2. Repeated Surgery

Another important pillar of therapy is, of course, surgery. It is necessary to thoroughly evaluate whether surgery makes sense for the patient at this point or not. Especially during initial diagnosis, persisting pain under adequate therapy is an important indication for surgery. In this case, surgery often proves to be very effective in reducing pain as well as central sensitization [19,20]. In case of a recurrence, however, the situation is different and the indication for repeated surgery should be carefully examined. It is highly recommended to read the surgical report and evaluate whether the endometriosis was removed. In order to assess the quality of an operation, it is helpful to check whether all endometriosis lesions were appropriately categorized and described and then removed/destroyed, e.g., by comparing the histological findings (do the number and size of the tissue samples correspond to the surgical procedure?). Especially in the case of peritoneal lesions, it is important to establish whether the first surgery was performed while the patient was on hormonal therapy, because, as Köhler et al. [21] have shown, endometriosis lesions are downstaged during hormonal therapy and vitreous active lesions, in particular, are not always visible. This is usually accompanied by a short-lived reduction in pain. In this case, repeated surgery can be considered (of course, without hormonal treatment). If it can be assumed, however, that the first surgery took place under ideal conditions, the patient should definitely be asked whether she experienced a reduction in pain. Repeated surgery can also be considered if the first surgery was performed more than two years ago and the symptoms are returning under hormonal therapy. In this case, however, central sensitization should also be considered and a change in hormonal therapy as well as multimodal concepts are recommended as first steps. In our own patient population, 7% underwent re-laparoscopy, and only two patients who declined hormonal therapy due to depression and the desire to have children experienced a recurrence of peritoneal endometriosis. One patient suffering from pronounced adenomyosis requested a hysterectomy, after which she was pain-free. Another patient had an isthmocele in which Nabothian cysts had developed, causing her pain (also uterine-related). In three women, no pain correlation could be found. Of the patients, 25% do not benefit from first-line laparoscopy and the proportion is even higher in chronic pain patients. Thus, we recommend further developing multimodal concepts instead of performing a second laparoscopy.

### 3.3. Analgesics

In case of persistent pain or intense discomfort, the principles of pain therapy should be discussed together with hormonal therapy. Painkillers cannot directly influence endometriosis lesions, but they inhibit the release of pain neurotransmitters and thus also have an anti-inflammatory effect. They intervene in cases of acute nociceptive pain. It is interesting that many patients are also reluctant to take painkillers. They are concerned/afraid that painkillers are dangerous and not healthy for the body. Many patients consciously try to endure the pain or only take painkillers “if there is no other way”. We recommend assessing the pain level according to the visual analogue scale (VAS) and asking patients whether they are bedridden and unable to work. Assessing the intensity of pain is very important. The patient should be encouraged to keep a pain diary so that all ups and downs can be recorded and evaluated. If patients do take painkillers, the product and dosage should be registered, as many patients make mistakes when taking them. Over-the-counter analgesics, for example, are sometimes extremely overdosed. When specifically asked about this, many women state that they have to take more and more painkillers because they frequently suffer from therapy-resistant pain (mostly women who do not receive hormonal therapy and whose cyclical complaints occur monthly). This is why basic information and advice concerning the purpose of pain therapy and the application regimen is obligatory. According to WHO recommendations, non-steroidal analgesics are part of first-line pain therapy. It is recommended that these be taken early on during the days of pain—in low doses, but regularly and following a fixed schedule. Waiting until the pain is no longer bearable is not advisable here, since it only means that very high doses must then be taken before the pain subsides. Ibuprofen, naproxen or metamizole are suitable analgesics. When taking them, it is advisable to also think about gastroprotection. One could start with an intake of ibuprofen 400 mg 1-1-1, which can then be increased to 3 × 800 mg. If necessary, one could add metamizole, i.e., in the form of drops. Unfortunately, there are very few studies investigating analgesics as pain medication for endometriosis. Only naproxen and ibuprofen have been the subject of some studies for the treatment of dysmenorrhea [22,23]. The WHO’s recommendations for pain management and its step-by-step approach (“pain ladder”) should thus be followed. The smooth muscles of the intestinal walls may also be another source of pain. In addition to analgesic therapy, the administration of Buscopan and high doses of magnesium seem to be useful in this case. According to several reports, cannabidiol (CBD) oil (10%) also seems to have a positive effect (freely available via the internet). Many patients report that using cannabis has a positive impact. The intake of tetrahydrocannabinol (THC) oil might also be an option, but has to be prescribed by the pain therapist. In individual cases, this has proved to be helpful [24]. The prescription of opiates should be strongly indicated and be at the discretion of the pain therapist. Overall, however, opiates do not seem to have a very strong effect on endometriosis-related pain, because opiate receptors are upregulated in the case of acute inflammatory pain, but not in chronic pain. The risk of becoming addicted is another reason why one should be extremely cautious with regards to opiates [25]. In cases of severe burning pain, a neuropathic cause should be taken into account, which means that gabapentin and pregabalin could be administered as well. Doloxetine and amitriptyline may also be used as co-analgesics. If these measures do not provide satisfactory pain control, the patient should be referred to a pain center.

### 3.4. Musculoskeletal System/Pelvic Floor

It is perfectly understandable and has been well-documented that pain can lead to poor posture and can affect the entire musculoskeletal system. There is extensive data on changes that occurred in patients with chronic pelvic floor pain. Significant changes are especially apparent in the patients’ tone, strength, stamina and their ability to coordinate and relax muscles [26,27]. If pain sensitization increases over the course of the disease, this will also lead to higher muscle tension, and changes in the pelvic floor involving hypertonic muscle groups will then further trigger the pain (myofascial trigger points). One method of therapy that alleviates pelvic floor dysfunction and its effects is electrostimulation. Intravaginal electrostimulation has already shown promising results in vaginismus and other pain disorders. In addition, Mira et al. have demonstrated in a study that therapy with transcutaneous electrical nerve stimulation (TENS) significantly reduces pain [25]. Patients should also perform relaxation exercises to relieve pelvic floor pain on their own and integrate them into their daily routine. Suggestions for exercises can be found here: https://www.youtube.com/watch?v=Auca88tmUu8; https://www.pelvicexercises.com.au/pelvic-floor-muscle-tension-article/; https://www.pelvicpain.org.au/our-mission/ (accessed on 10 January 2022). Endometriosis patients can also be prescribed physiotherapy or manual therapy under the indication of lumbar spine syndrome. Regarding the pelvic floor, it must be emphasized that dyspareunia plays a significant role in the patient’s suffering. It is therefore important to discuss this topic with the patient, and if necessary with her partner, and to point out the possibility of seeking sexual medical counseling. In this case, pelvic floor relaxation exercises, TENS applications as well as yoga [28] and osteopathy are helpful, too. Since patients often suffer from deep dyspareunia, devices that reduce the depth of penetration can also be useful (https://cucabylinda.com/product/ohnut-set/, accessed on 10 January 2022).

### 3.5. Diet

It is interesting to note that an ever-increasing role is attributed to the patients’ diet when it comes to the development and progression of endometriosis. More and more patients observe that their unspecific intestinal complaints, which they themselves have never considered to be a “gynecological problem”, follow a cyclical pattern. This phenomenon is known as “endo belly”, which means that the patients’ belly is severely bloated (Figure 3). This causes not only pain and feeling unwell, but it is also accompanied by functional bowel emptying disorders (constipation/diarrhea), food intolerances and pain before defecation. Unfortunately, the number of studies on evidence-based dietary recommendations is very limited [29]. Given the complexity of the disease, however, this is not surprising [27]. In general, patients show a benefit with a high-fiber diet with plenty of fruits and vegetables, and a reduced consumption of sugar and animal products are recommended [30]. One can find various recommendations for an “endometriosis diet” on the internet, which give the impression that the effect of a certain diet is proven (Endometriosevereingung Austria (EVA), Association for Endometriosis Austria). However, some of these recommendations are not based on empirical studies. In our experience, a vegan diet without industrial sugar and gluten often leads to a significant improvement in symptoms (unpublished data). Especially milk products and pork might be sources of inflammatory substances such as prostaglandins, so good-quality meat such as chicken (white meat) and fish might be ok occasionally. It is assumed that parts of the intestinal mucosa/wall are also subject to cyclical changes; the microbiome, but also chronic inflammation, may be involved in this. This phenomenon needs to be examined in further studies [31]. Currently, there are various approaches to build up and stabilize the intestinal flora with probiotics. It is generally known that a pro-inflammatory environment gives rise to free radicals and leads to oxidative stress. Thus, an anti-inflammatory diet could be useful. A prospective study has shown, for example, that the intake of fish oil (omega-3 unsaturated fatty acids) reduced the risk of EM recurrences [32]. The ratio of unsaturated fatty acids to saturated fatty acids was important in this context [32]. Vitamin C, E and A, selenium and zinc also have an antioxidant effect. In addition to its anti-inflammatory effect, vitamin D is also said to have an antiproliferative and immunomodulatory effect, and in one study even caused a reduction in pain [33,34,35], probably due to a reduced histamine release. Several compositions of anti-inflammatory substances are already freely available for the supportive therapy of endometriosis.

### 3.6. Psyche

Another important factor in the development and experience of the disease is, of course, the psyche. Various studies have shown that the health-related quality of life of women with endometriosis is significantly lower compared to women who do not suffer from endometriosis [36]. This is particularly evident in infertile patients with endometriosis compared to those who do not have endometriosis [26]. Moreover, women with endometriosis also show increased psychopathological symptoms. International studies indicate that 15% to 87% of women with endometriosis suffer from depressive symptoms and 29% to 88% have anxiety disorders [37,38,39]. Detailed analyses suggest that the cause of psychological distress is not endometriosis as such, but rather the pain associated with endometriosis, especially when it is chronic. Endometriosis patients with severe pain are significantly more likely to show depressive symptoms and anxiety disorders than those without pain [40,41]. This is why psychological support is essential and should be discussed and initiated. Unfortunately, the provision of psychological support is very limited due to the lack of therapists. Still, first steps can be taken in the context of rehabilitation measures. It is important to find personal strategies for coping with the disease. Pain education courses to develop pain management strategies and learning various techniques such as progressive muscle relaxation, autogenic training, relaxation exercises, yoga, chi gong, creative therapies, hypnosis, etc., are also extremely useful and should be integrated into the patient’s daily routine. A multimodal therapy can also include complementary methods. With regards to acupuncture in particular, a number of studies have proven its positive effects. It is assumed that the release of endogenous opioids (pain reduction) and endogenous cortisol (anti-inflammatory), deactivation of brain areas linked to pain perception and local effects such as adenosine release and local blood flow modulate the pain [42]. Several studies have confirmed an improvement of endometriosis-associated pain [43]. Local measures such as Botox injections [44] and neural therapy [45] can also be applied on an individual level. Homeopathy, balneotherapy, traditional Chinese medicine, etc., are also of value and should not be underestimated as supportive measures. Rehabilitation: In Germany, we are in the very comfortable situation that there are certified rehabilitation clinics that care for endometriosis patients. One can apply for rehabilitation treatment following an inpatient stay or as a rehabilitation measure. To date, this option is not being fully utilized. The program offered by these clinics is designed for endometriosis patients and provides disease-specific information, psychological support, pain education courses and coping strategies, intensive physiotherapy and many other treatments. Patients can also get advice from social services regarding job-specific measures or applying for a degree of disability.

### 3.7. Pain Complex Therapy

All of these steps are components of a multimodal therapy. If it is not possible to establish an outpatient network for the multimodal therapy of an endometriosis patient, or if the measures are not effective, inpatient multimodal pain therapies offered by some pain centers are also an option. In this case, it is important to ensure that the gynecological aspects of the organically caused disease are also taken into account and that the main component of the multimodal pain therapy is not primarily a psychosomatic approach—a joint (gynecological and pain therapy) concept would be ideal. Our own experience has shown that this group of patients is highly motivated and that their “efforts” are rewarded by positive results. Endometriosis patients often have such a high level of suffering that they are open to all options that promise to reduce their pain.

## 4. Conclusions

Endometriosis-associated pain is caused by peripheral (activity of lesions and pain nerve fibers) and central sensitization. Our step-by-step approach was designed to help consider the various aspects of therapy. Central sensitization and its effects are often underestimated, but ultimately lead to patients slipping into a chronic pain syndrome that can gradually worsen. Secondary myofascial pain in the pelvis, in particular, significantly reduces the patients’ quality of life. When the pain becomes chronic, it can also overlap with other chronic pain syndromes such as irritable bowel syndrome (IBS) and painful bladder syndrome or vulvodynia. This is easily recognizable by the lowered pain threshold. In this case, multimodal therapy is even more important.

## Figures and Tables

**Figure 1 jcm-11-00467-f001:**
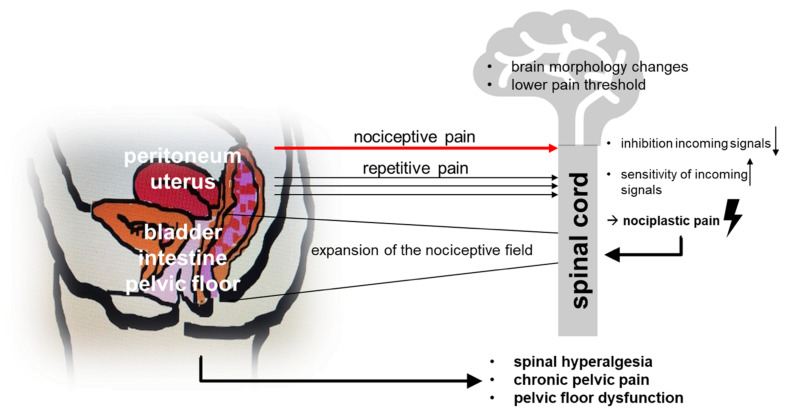
Pathophysiology of chronic pain disorders in women with endometriosis.

**Figure 2 jcm-11-00467-f002:**
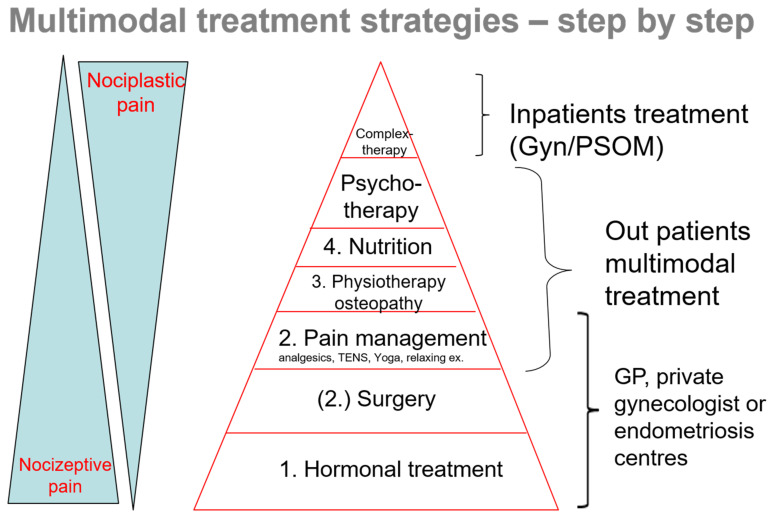
Multimodal treatment strategies—step-by-step: the basis forms hormonal treatment, surgery is optional, facultative is pain management in addition to physiotherapy/osteotherapy, nutrition and in more complex cases psychotherapy and gynecological/psychosomatic inpatient treatment (Gyn/PSOM) as interdisciplinary intervention.

**Figure 3 jcm-11-00467-f003:**
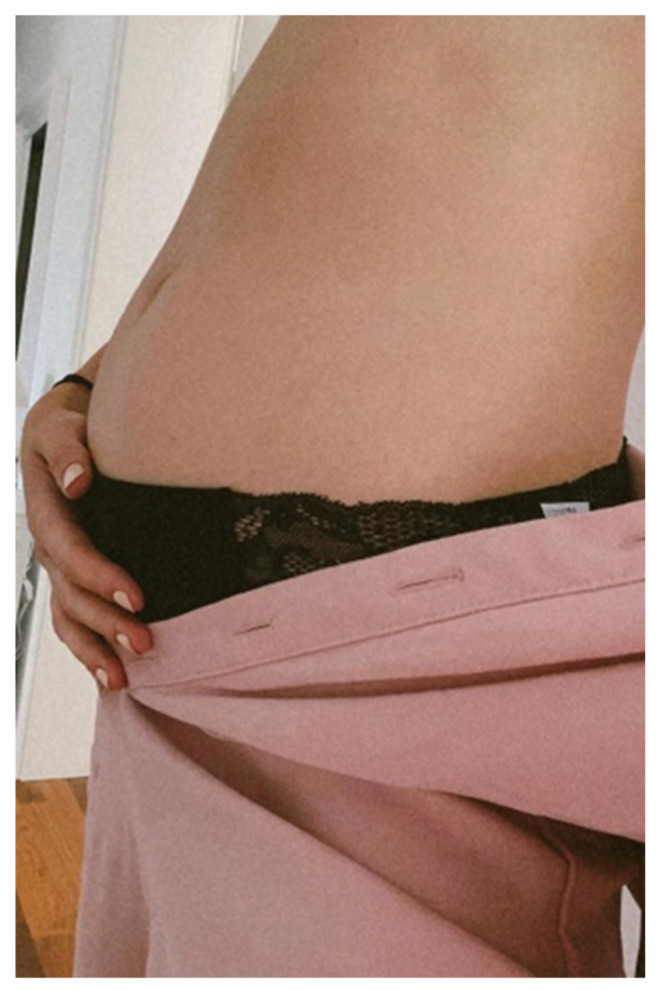
The endobelly: women with bowel emptying disorder.

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
