# Peer review of "Endometriosis, an Ongoing Pain—Step-by-Step Treatment"

_jcm, 2022, doi:10.3390/jcm11020467_

Round 1

Reviewer 1 Report

This article addresses important issues in the long-term treatment of endometriosis. The attention was devoted to the practical problems of recognizing endometriosis and treating symptoms without obtaining a final diagnosis, and the problem of standardization of care, which still leaves much to be desired even in highly developed countries. It is written in stylistically correct English, but improvement is occasionally useful. Nevertheless, there are also some substantive issues that need to be refined:
1. References are not formatted in alphabetical order or in the order in which they appear in the manuscript.
2. The considerations in the introduction that compares endometriosis with diabetes are rather redundant. There are certainly big differences depending on the social group and geographic location. However, if such theses are made, they should be properly justified by scientific research - there is no reference here. The presentation of the complex mechanism of diabetes (we have different types of diabetes and diabetes in various clinical situations, e.g. during pregnancy) also seems too simplified and incorrect.
3. It has been argued that progestin monotherapy is preferable. In contrast, both the WES guidelines and ESHRE state that OCP and progestins are equivalent first-line treatments. OCP in cyclic and continuous therapy can be better tolerated and has fewer side effects (e.g. psyche, bones, breakthrough bleeding). Off-label cannot be an argument, because in the case of endometriosis, the IUD is also off-label, and the treatment principle is to be up-to-date medical knowledge, effectiveness and safety, and not the regulations of pharmaceutical companies.
4. There is an incomprehensible link in line 158: (https://www.msn.com/de-de/gesundheit/medizinisch/nimmt-frau-heute-noch ...). If needed, such links should be given in its entirety and as a reference in the bibliography section.
5. Many abbreviations have been used in various places that are not developed anywhere: IBS, CBD, THC. Although the reviewer understands the abbreviations, they are not universally understood and the full name must be provided the first time it is used in the text.
6. The part of the diet seems to be partially speculative. There are many pertinent comments regarding gluten, omega-3 fatty acids, and trans fatty acids. However, recommending a vegan (why not vegetarian?) diet is not justified in the literature. Many studies show no relationship between meat consumption and the development and treatment of endometriosis. Only finally did an NHS II study show a link between eating very large amounts of red meat (2 servings a day) and a slight increase in the risk of endometriosis. On the other hand, the elimination of meat from the diet is not confirmed by theoretical considerations, clinical trials and everyday medical practice.

Author Response

Reviewer 1

This article addresses important issues in the long-term treatment of endometriosis. The attention was devoted to the practical problems of recognizing endometriosis and treating symptoms without obtaining a final diagnosis, and the problem of standardization of care, which still leaves much to be desired even in highly developed countries. It is written in stylistically correct English, but improvement is occasionally useful. Nevertheless, there are also some substantive issues that need to be refined:

  1. References are not formatted in alphabetical order or in the order in which they appear in the manuscript.

changed

  1. The considerations in the introduction that compares endometriosis with diabetes are rather redundant. There are certainly big differences depending on the social group and geographic location. However, if such theses are made, they should be properly justified by scientific research - there is no reference here. The presentation of the complex mechanism of diabetes (we have different types of diabetes and diabetes in various clinical situations, e.g. during pregnancy) also seems too simplified and incorrect.

 thank you for this comment, I understand what the reviewer mean, but my intention was to highlight the different general education about the two diseases and the perceptual imige of the diseases in the public view. Of course diabetes is also a really complicated disease, so I did´t adress the different forms. However, both disaeses share a high incidence/prevalence but they have a different awareness. I added a reference.

  1. It has been argued that progestin monotherapy is preferable. In contrast, both the WES guidelines and ESHRE state that OCP and progestins are equivalent first-line treatments. OCP in cyclic and continuous therapy can be better tolerated and has fewer side effects (e.g. psyche, bones, breakthrough bleeding). Off-label cannot be an argument, because in the case of endometriosis, the IUD is also off-label, and the treatment principle is to be up-to-date medical knowledge, effectiveness and safety, and not the regulations of pharmaceutical companies.

I changed as recommended

  1. There is an incomprehensible link in line 158: (https://www.msn.com/de-de/gesundheit/medizinisch/nimmt-frau-heute-noch ...). If needed, such links should be given in its entirety and as a reference in the bibliography section.

thank you, you are right, the link is not working, I selected another one. Edited

  1. Many abbreviations have been used in various places that are not developed anywhere: IBS, CBD, THC. Although the reviewer understands the abbreviations, they are not universally understood and the full name must be provided the first time it is used in the text.

 you are complety right, changed

  1. The part of the diet seems to be partially speculative. There are many pertinent comments regarding gluten, omega-3 fatty acids, and trans fatty acids. However, recommending a vegan (why not vegetarian?) diet is not justified in the literature. Many studies show no relationship between meat consumption and the development and treatment of endometriosis. Only finally did an NHS II study show a link between eating very large amounts of red meat (2 servings a day) and a slight increase in the risk of endometriosis. On the other hand, the elimination of meat from the diet is not confirmed by theoretical considerations, clinical trials and everyday medical practice.

You are right good data are missing, and have to be evaluated in the near future. however the new study of Krabbenborg et al., confirmed that food changes improve the quality of life. In our experince the vegan diet with reduced sugar and gluten showed significant improvement of bowel related symptoms and pain reduction, too (unpublished data), I would like to share this experience. I edited the paragraph.

Reviewer 2 Report

The author summarizes the treatment for endometriosis by including the author’s opinion. This manuscript is well written; however, there are several points to be addressed. The reviewer’s comments are listed below.

There are numerous mistakes in the reference numbers. References must be serially numbered and should be arranged in increasing order of numbers quoted in the text. Please check carefully and revise accordingly.

There are several run-on paragraphs in the manuscript. Too many lines in a paragraph is difficult to read. Please revise.

The reviewer thinks the author may discuss the role of an aromatase inhibitor in the treatment of endometriosis. A previous systematic review of eight studies concluded that aromatase inhibitor treatment, with or without progestin, COC, or GnRH analog, significantly reduced pain compared with GnRH analog alone.

Nerve transection procedures, including laparoscopic uterosacral nerve ablation (LUNA) and presacral neurectomy (PSN), have been used to treat pelvic pain caused by endometriosis. The author may discuss the role of LUNA and PSN in the surgical treatment of endometriosis.

Minor comments

Line 170: Please cite the previous study.

Line 204: Please clarify the medical cost of elagolix.

Please do not use abbreviations without definition (including tables). Once an abbreviation is defined in the main text, thereafter can it only be used throughout the manuscript.

Line 182: POPs, Line 200: HRT, Line 402: IBS

Author Response

Reviewer 2

The author summarizes the treatment for endometriosis by including the author’s opinion. This manuscript is well written; however, there are several points to be addressed. The reviewer’s comments are listed below.

There are numerous mistakes in the reference numbers. References must be serially numbered and should be arranged in increasing order of numbers quoted in the text. Please check carefully and revise accordingly.

revised

There are several run-on paragraphs in the manuscript. Too many lines in a paragraph is difficult to read. Please revise.

revised

The reviewer thinks the author may discuss the role of an aromatase inhibitor in the treatment of endometriosis. A previous systematic review of eight studies concluded that aromatase inhibitor treatment, with or without progestin, COC, or GnRH analog, significantly reduced pain compared with GnRH analog alone.

You are right, I ve forgotten and now I added this. We have also good experience with it.

Nerve transection procedures, including laparoscopic uterosacral nerve ablation (LUNA) and presacral neurectomy (PSN), have been used to treat pelvic pain caused by endometriosis. The author may discuss the role of LUNA and PSN in the surgical treatment of endometriosis.

Thank you for this hint, but I am carefully with this recommendation, because of limited data. And this option is in my opinion something for really well trained centers with outstanding experience in this field. In generally I don´t want to open the discussion about surgical procedures in case of recurrent disease because this is very complex and this topic has to be regarded in an own article and, but I will keep in mind, would be good to evaluate this in the near future.  

Minor comments

Line 170: Please cite the previous study. No studies available about bioidentical hormones and endometriosis, the hint is based on patient reports

Line 204: Please clarify the medical cost of elagolix. Added  (1050 Euros für 28 tbaletts), https://de.thesocialmedwork.com/shop/women-s-health/endometriosis/orilissa-elagolix

Please do not use abbreviations without definition (including tables). Once an abbreviation is defined in the main text, thereafter can it only be used throughout the manuscript.

Line 182: POPs, Line 200: HRT, Line 402: IBS, Thank you for this hint, it´s changed now

Round 2

Reviewer 1 Report

Most of the comments were positively taken into account.
I still have some comments:
1. In Fig. 1 "multimodale" must be changed to multimodal and "Endometriosis" to endometriosis. I also do not fully understand the numbering of the grades. Why is (2) in parentheses? Maybe it needs to be changed or clarified under the figure.
2. The dietary sub-paragraphs may remain in their present form. There is indeed no good research on this and that is to be regretted. Perhaps this should be emphasized more. The experience from the endometriosis center run by myself shows that a gluten-free and dairy-free (casein) diet is beneficial for many patients. Unfortunately, we do not have hard evidence on this. We are not, however, convinced that the meatless diet plays a role, but these are merely personal observations.
After changing the above-mentioned typos may be published.

Author Response

Reviewer 1

Most of the comments were positively taken into account.
I still have some comments:
1. In Fig. 1 "multimodale" must be changed to multimodal and "Endometriosis" to endometriosis. I also do not fully understand the numbering of the grades. Why is (2) in parentheses? Maybe it needs to be changed or clarified under the figure.

Thank you for this hint, edited

  1. The dietary sub-paragraphs may remain in their present form. There is indeed no good research on this and that is to be regretted. Perhaps this should be emphasized more. The experience from the endometriosis center run by myself shows that a gluten-free and dairy-free (casein) diet is beneficial for many patients. Unfortunately, we do not have hard evidence on this. We are not, however, convinced that the meatless diet plays a role, but these are merely personal observations.
    After changing the above-mentioned typos may be published.

done

Reviewer 2 Report

The author revised the manuscript according to my previous comments.

Author Response

Thank you for your comment.